# Novel Complexes of 3-[3-(1*H*-Imidazol-1-yl)propyl]-3,7-diaza-bispidines and β-Cyclodextrin as Coatings to Protect and Stimulate Sprouting Wheat Seeds

**DOI:** 10.3390/molecules27217406

**Published:** 2022-11-01

**Authors:** Altynay B. Kaldybayeva, Valentina K. Yu, Aigul E. Malmakova, Tamara Li, Assel Yu. Ten, Tulegen M. Seilkhanov, Kaldybay D. Praliyev, Kenneth D. Berlin

**Affiliations:** 1Laboratory of Chemistry of Synthetic and Natural Medicinal Substances, A.B. Bekturov Institute of Chemical Sciences, 106 Sh. Ualikhanov St., Almaty 050010, Kazakhstan; 2Department of Chemistry and Technology of Organic Substances, Natural Compounds and Polymers, Al Farabi Kazakh National University, 71 Al-Farabi Ave, Almaty 050040, Kazakhstan; 3Laboratory of Cell Engineering, Institute of Plant Biology and Biotechnology, 45, Timiryazev Str., Almaty 050040, Kazakhstan; 4Laboratory of the Engineering Profile of NMR Spectroscopy, Sh. Ualikhanov Kokshetau University, 76, Abay St., Kokshetau 020000, Kazakhstan; 5Department of Chemistry, Oklahoma State University, Stillwater, OK 74078, USA

**Keywords:** 1-[3-(1*H*-imidazol-1-yl)propyl]piperidine, 3-[3-(1*H*-imidazol-1-yl)propyl]-3,7-diazabicyclo[3.3.1]nonanes, β-cyclodextrin, growth-stimulating activity, activity–complex correlations

## Abstract

We report the syntheses and characterization of novel 3,7-bicycl[3.3.1]bispidines possessing an imidazolpropyl group attached to N-3, and at N-7 a Boc group, as well as a benzoylated-oximated group at C-9. These compounds were complexed with β-cyclodextrin [β-CD] and evaluated as seed protectors of selected wheat seedlings. Using strong acid, condensations of N-substituted piperidones with the appropriate imidazolpropyl groups at N-3 and N-7 led to bispidinones **6** and **7**. These intermediates were reduced to the corresponding 3,7-diazabicyclo[3.3.1]nonane targets. The oxime at C-9 was benzoylated to yield **13**. Heating these 3,7-diazabicyclo[3.3.1]nonanes in ethanol with β-CD generated the complexes required. We investigated the ability of such complexes as coatings on seedlings to protect and stimulate growth of three varieties of wheat, namely Kazakhstanskaya-10, Severyanka, and Miras. The complex of 3-[3-(1*H*-imidazol-1-yl)propyl]-7-(3-methoxypropyl)-3,7-diazabicyclo[3.3.1]nonane (**2**) promoted growth in the root systems of all three wheat varieties by more than 30% in Kazakhstanskaya-10, 30% in Severyanka and 8.5% in Miras. A complex of 3-Boc-7-[3-(1*H*-imidazol-1-yl)propyl]-3,7-diazabicyclo[3.3.1]nonane (**9**) increased both shoot and root length in only the Severyanka variety. The complex of 3-(3-butoxypropyl)-7-[3-(1*H*-imidazol-1-yl)propyl]-3,7-diazabicyclo[3.3.1]nonane (**11**) stimulated both shoot growth (0.8%, 12.3%, 13.5%) and root growth (12.3%, 9.4%, 21.7%) in all three varieties of wheat, respectively. The nature of substituents on the bispidine affect the activity. Solid complexes (1:1) were generated as powders which melted above 240 °C (dec) and were characterized via elemental analyses as 1:1 complexes.

## 1. Introduction

There is a current urgency to identify new protectors/regulators for coating seeds [1,2,3]. In modern technologies, regarding crop production, the practical importance and need of seed protectors/regulators are critical for the acceleration of growth and increased yields of many agricultural crops [4,5,6,7,8,9,10,11,12,13]. The selection of such regulators should be environmentally friendly and economically profitable to increase crop yields and demonstrate more fully the potential of plant organisms [14,15]. Preliminary work revealed the potential for stimulating seed germination with a few mono- and bicyclic systems with an attached imidazole group [16,17,18]. In one example, an inclusion complex of β-cyclodextrin with 1-[3-(1*H*-imidazol-1-yl)propyl]piperidine (**1****-**Figure 1) stimulated the germination of seeds of soybeans, corn, wheat, barley, and onions. One other example that appeared in the literature was 3-[3-(1*H*-imidazol-1-yl)propyl]-7-(3-methoxypropyl)-3,7-diazabicyclo[3.3.1]nonane complex (**2**) [15], which was used as a standard in the present work. The use of β-CD for moisturizers and solubilizers is known [19]. A major objective of the present work was to incorporate hygroscopic units to attract water to the seeds and retain a hydrocarbon portion to protect the seeds from pathogens.

## 2. Results

### 2.1. Synthesis of 3,7-Diazabicyclo[3.3.1]nonane Derivatives

Based, in part, on preliminary work [16,17,18], it was reasoned that lengthening an alkoxyalkyl chain at the nitrogen atom on a piperidine ring could provide useful coatings for seeds. Consequently, the syntheses of selected 3,7-diazabicyclo[3.3.1]nonanes containing a piperidine ring were initiated.

The reaction of 3-(1*H*-imidazol-1-yl)propan-1-amine (**3**) and paraformaldehyde with N-substituted piperidones **4** and **5** and *N*-3-aminopropylmidazole under acid conditions led to bispidinones **6** and **7** (65% and 68% yields, respectively—Figure 1). Characterization of intermediates **8** and **10** and the complexes was achieved with IR analyses. NMR spectral analyses, specific chemical tests, and elemental analyses. A few examples of the use of β-CDs for encapsulating large molecule are known [20].

The IR spectra of 3,7-diazabicyclo[3.3.1]nonan-9-ones **6** and **7** contain signals for the carbonyl group (at C-9) in the region of 1702 cm^−1^ and 1734 cm^−1^, respectively. Absorption at 1651 cm^−1^ was attributed to the carbonyl group in the ester function of ketone **6**. In the ^13^C-NMR spectrum, the lowest-field signals at 213.7 ppm (**6**) and at 214.4 ppm (**7**) were attributed to the ketone carbons (C-9), respectively. Carbon signals at C-2 and C-4 (**6**), as well as for C-6 and C-8 (**7**), appeared at 47.4 ppm, 53.7 ppm (**6**) and at 66.9 ppm 68.6 ppm, respectively (**7**). The carbon atoms of the imidazole group resonated at 119.9–137.8 ppm. The carbonyl group at 154.6 ppm was assigned to the carbonyl carbon of the carboxyl group in **6**. The ^1^H-NMR spectra of **6** and **7** displayed two proton signals at the bridgehead carbons (H-1,5) in the range of 2.90–3.51–1.68 ppm.

The ^1^H-NMR signals of the axial protons of 3,7-diazabicyclo[3.3.1]nonan-9-one (**7**) at C-2,4 and C-6,8 were multiplets at 2.18–2.33 ppm. The resonance signals of the equatorial protons at C-2,4 and C-6,8 were in the lower-field of the spectrum (2.90–3.01 ppm). In bispidinone (**6**) with the carboxylate group, signals for axial and equatorial ring protons were at 2.05–2.46 ppm and 3.05–3.51 ppm for H-2,4, and at 3.88–3.95 ppm and 4.24–4.41 ppm for H-6,8.

3-Boc-7-[3-(1*H*-imidazol-1-yl)propyl]-3,7-diazabicyclo[3.3.1]nonane (**8**) (52%) and 3-(3-butoxypropyl)-7-[3-(1*H*-imidazol-1-yl)propyl]-3,7-diazabicyclo[3.3.1]nonane (**10**) (40%) were obtained by reduction of the corresponding bicyclic ketones **6** and **7**. The absence of the carbonyl IR band in **8** and **10** confirmed that reduction had occurred. The composition and structure of compounds **8** and **10** were further supported from IR, ^1^H NMR, and ^13^C-NMR spectral analyses. When comparing the ^13^C-NMR spectra of compounds **8** and **10** with the spectra of the starting ketones **6** and **7**, the appearance of a triplet for H-9 in the ^1^H-NMR spectrum and the ^13^C-NMR signals at 35.0 ppm and 35.1 ppm, respectively, for C-9 substantiated the conversion of ketones **6** and **7** to the corresponding **8** and **10** derivatives. Signals for 8 and 10 were visible in the region of 1.53 ppm and 1.88 ppm signals for the H-9 methylene protons, respectively. The signals for H-1,5 protons in 8 and 10 were shifted upfield (1.32 ppm and 1.89 ppm, respectively) in comparison to the initial ketones.

It was reasoned that providing a group at C-9 which had H-bonding capabilities might increase moisture attraction capability in the complexes. Therefore, oximation was initiated with of 3-(3-butoxypropyl)-7-[3-(1*H*-imidazol-1-yl)propyl]-3,7-diazabicyclo[3.3.1]nonan-9-one (**7**) and gave, the corresponding oxime **12** (49%). The appearance in the IR spectrum of bands in the region of 1506 cm^−1^ and 3200 cm^−1^ for **12** indicated the formation of an oxime and were assigned, respectively, to the C=N and O-H groups. The most intense band in **13** was the benzoyloxy carbonyl group (1723 cm^−1^). Oxime **12** was acylated with benzoyl chloride in absolute benzene to give the benzoyloxime **13** (45%). The ^13^C-NMR spectra of oxime **12** and its ester **13** showed resonances for C-9 at 159.5 ppm and 169.1 ppm, respectively. A second ^13^C-NMR signal at 166.1 ppm identified the carbonyl carbon for the acyl residue. In the ^1^H-NMR spectra, H-1.5 protons were observed as separate signals at 2.71 ppm and 2.95 ppm in oxime 12 and at 3.23 ppm and 3.30 ppm for the N-benzoyl derivative. A weak signal at 6.82 ppm was assigned to the proton of the NOH group (**12**). Signals for the protons of the phenyl ring (**13**) were observed at 7.49–7.94 ppm.

In the (1*H*-imidazol-1-yl)propyl) substituent, the ^13^C atoms of the imidazole fragment appeared at 119.9–138.80 ppm in all systems. Signals for propyl carbons attached to nitrogen atoms were observed in the range of 44.0–50.5 ppm, while the carbons between them gave a signal at 28.5–31.8 ppm. The ^1^H-NMR spectrum showed signals at 6.81–7.57 ppm from the protons of the imidazole ring. Protons of methylene groups attached to nitrogen atoms gave signals in the range of 3.05–4.38 ppm. The ^1^H-NMR signals at 1.55–1.78 ppm were assigned to the protons of the second CH_2_ group.

The ^13^C-NMR signals for O-C at 28.5 ppm and at 28.3 ppm for the CH_3_ group were observed for the carboxylate, respectively, in bispidinone (**6**) and bispidine (**8**). The protons of the methyl group of the *t*-butyl fragment appeared at 1.22–1.48 ppm. NMR spectra of products **7, 10, 12,** and **13** had the expected full set of signals for atoms of carbon and hydrogen of the butoxypropyl substituent.

The reaction products (**7, 8**, **10**, **12**, **13**) were viscous oils soluble in organic solvents. The β-CD complexes were solids and were used to study the biological properties of the bispidines as stimulators/growth factors. The solid complexes (1:1) of **9**, **11**, and **14** melted above 240 °C with decomposition.

### 2.2. Growth-Stimulation Activity

There is a major need for materials to coat seeds as noted recently [1,2,3]. One method of increasing grain yield is the preparation of seed material in terms of its disinfection and protection of the seedlings from external pathogenic factors. Moreover, increasing resistance to pathogens is critical, as is the stimulation of plant growth and development, both during seed germination and during the growing season. In the present work, it was possible to achieve positive results due to seed dressing and pre-sowing treatment with growth regulators. This can lead to the mobilization of potential genotypically determined plant capabilities and, ultimately, to increased yields. Figure 1 illustrates the syntheses and the variety of 3,7-diazabicyclo[3.3.1]nonanes and complexes prepared with β-CD. Compounds **8**, **10**, and **13** readily formed complexes **9**, **11**, and **14** with β-CD, respectively.

The synthetic “assembly” of piperidine and imidazole fragments into one molecule led to specific biologically active substances as growth protectors/regulators of specific wheat seedlings. As can be seen from the test results, the composition of the ring structure of the six-membered azaheterocycle (piperidine unit in 3,7-diazabicyclo[3.3.1]nonane) in complexes with β-CD led to a change in bioactivity (increased in certain categories except for samples **1** and **14**). The presence of the alkoxyalkyl substituent at the position of 7 in the 3,7-diazabicyclo[3.3.1]nonane ring (**2, 11**) strongly enhanced the bioaction on the growth of wheat seedlings while the shoot growth was slightly inhibited in **2**. However, the root length increased. Only example **14** in the 3,7-diazabicyclo[3.3.1]nonanes complex systems involving substitution at C-9 gave an inferior effect on plant development in each variety of wheat. Thus, the nature of the substituent has a dramatic affect for maximum results.

## 3. Discussion

Figure 2 below demonstrates the results from experiments conducted with the various wheat varieties in terms of seedling development and growth of plant height (shoot) and root length. The treatment process consisted of pre-sowing seeds with 0.01% aqueous solutions for 10 days. Complexes **1** and **2** were tested at the Institute of Plant Biology and Biotechnology [16] and were used as models for the growth-regulating ability. Interestingly, pre-sowing seed did not reveal noticeable differences in germination of wheat of the three varieties examined. the stem and roots in all three wheat varieties. With **11** (7-butoxypropyl analog), growth was stimulated in both the stem and length of the roots in all three varieties of wheat. Surprisingly, shoot growth in the three wheat varieties was only slightly inhibited by **2** by about 3–5%. Interestingly, experiments revealed that the pre-sowing treatment of seeds with samples **1, 2, 9,** and **14** resulted in the plant height being reduced from 0.36% to 24.3%, and the root length decreased from 2.6% to 31.2% (**1, 9, 14**). Sample **14** inhibited the growth of the shoot and roots in all three wheat varieties. Table 1 shows the mean and standard deviation for each sample compared to the control. It clear that a substituent at C-9 is critical for root and shoot growth.

Figure 3 illustrates measuring the growth of plant height (shoot) and root length in the various wheat varieties. The length of the roots was increased remarkably by complex **2** in all three wheat seedlings, experiencing growth of 31–33% (Table 1).

## 4. Materials and Methods

### 4.1. Chemical Experimental Part

#### 4.1.1. Reagents and Equipment

The reaction of 3-(1*H*-imidazol-1-yl)propan-1-amine (**3**) and paraformaldehyde with N-substituted piperidones **4** and **5** with *N*-3-aminopropylimidazole under acidic conditions led to bispidinones **6** and **7** (65 and 68% yields, respectively-Figure 1). Characterization of the intermediates **8** and **10** and the complexes was achieved with spectral analyses (IR, NMR), specific chemical tests, and elemental analyses. The use of β-CDs for encapsulating large molecule is known [20].

The course of the synthetic reactions and the individuality of the compounds generated were monitored by TLC on Al_2_O_3_ with the appearance of spots detected with iodine vapor. IR spectra were recorded on a Nicolet-5700 spectrometer. ^1^H and ^13^C-NMR spectra were recorded on a JNM-ECA Jeol 400 spectrometer (frequencies 399.78 and 100.53 MHz, respectively) using DMSO-*d*_6_ as solvent. Chemical shifts were measured relative to residual protons or carbons of deuterated dimethyl sulfoxide. The initial reagents were obtained from Sigma-Aldrich: 3-(1*H*-imidazol-1-yl)propan-1-amine (**3**) and 1-Boc-piperidin-4-one (**4**), but 1-(3-butoxypropyl)piperid-4-one (**5**) was synthesized [16]. The NMR signals found for the bispidines and bispidinones were close to related but different systems [21,22,23,24,25,26,27]. Elemental analyses were performed on new compounds by Atlantic Microlabs (Norcross, GA, USA). The IR and NMR spectra for the synthesized compounds are presented in the Appendix A.

#### 4.1.2. Syntheses of Bispidine-9-ones (Compounds **6**–**7**)

*3-Boc-7-[3-(1H-imidazol-1-yl)propyl]-3,7-diazabicyclo[3.3.1]nonan-9-one* (**6**). A system composed of a three-necked flask equipped with a stirrer, reflux condenser, and dropping funnel, to which was added 40 mL of methanol that had been deoxygenated with nitrogen. After 30 min, a mixture of 3-(1*H*-imidazol-1-yl)propan-1-amine (**3.** 6 mL, 50 mmol), paraformaldehyde (6.02 g, 201 mmol), concentrated hydrochloric acid (2.65 mL), and glacial acetic acid (3.77 mL) was added with stirring for 15 min under nitrogen. An addition of 1-Boc-piperidin-4-one (10 g, 50 mmol, **4**) in glacial acetic acid (4 mL) in methanol (15 mL) was added dropwise. After heating the reaction mixture for 10 h at 60–65 °C, a second equivalent of paraformaldehyde was added, and the solution was maintained for 12 h at the same temperature. During the entire reaction, the mixture was purged with nitrogen. The solvent was evaporated, and the residue was dissolved in water (80 mL). Extraction of the neutral products with diethyl ether gave two layers, and the aqueous layer was made alkaline with NaOH. Upon cooling the mixture (at pH 12), the product was extracted with chloroform. The extract was dried (MgSO_4_), and the solvent was evaporated. The resulting solid was purified by column chromatography over Al_2_O_3_ with the eluent being benzene-isopropanol (6:1). The ketone 3-Boc-7-[3-(1*H*-imidazol-1-yl)propyl]-3,7-diazabicyclo[3.3.1]nonan-9-one (**6,** 11.43 g, 65%) was obtained and possessed an Rf 0.56 (Al_2_O_3_, eluent-benzene-isopropanol = 6:1), mp 82–85 °C.

Calculated for C_18_H_28_N_4_O_3_: C, 62.05; H, 8.10; N, 16.08.

Found: C, 62.07; H, 8.07; N, 16.06

IR spectrum cm^−1^: 1702 (C=O), 1651 (COO), 1164 (C-O-C).

^1^H NMR spectrum, ppm: *bicyclononan-9-on:* 2.05–2.46 (2 H, H-2_ax_,4_ax_), 3.05–3.51 (4 H, H-1,5, 2 _eq_,4 _eq_), 3.88–3.95 (2 H, H- 6_ax_,8_ax_), 4.24–4.41 (2 H, H- 6_eq_,8 _eq_). (*1H-imidazol-1-yl)propyl**:* 6.81–6.83, 7.03–7.08, 7.47–7.54 (3 H, CH_imidazol_). Side chains: 1.71–1.78 (2 H, NCH_2_CH_2_CH_2_N); 3.05 3.51 (4 H, NCH_2_CH_2_CH_2_N*). Boc*: 1.22–1.32 (9 H, CH_3_).

^13^C NMR spectrum ppm: bicyclononan-9-on: 49.8 (C-1,5), 47.4, 53.7 (C-2,4,6,8), 213.7 (C-9). *(1H-imidazol-1-yl)propyl:* 119.9, 128.7 137.8 (CH_imidazol_). Side chains: 31.5 (CH_2_CH_2_CH), 50.5 (CH_2_CH_2_CH_2_). *Boc:* 28.5 (CH_3_), 79.5 (O-C), 154.6 (C=O).

*3-(3-Butoxypropyl)-7-[3-(1H-imidazol-1-yl)propyl]-3,7-diazabicyclo[3.3.1]nonan-9-one* (**7**). In a three-necked flask equipped with a stirrer, reflux condenser, and addition funnel, was placed 40 mL of methanol which was then deoxygenated under nitrogen. After 30 min, a mixture was generated by adding 3-(1*H*-imidazol-1-yl)propan-1-amine (**3**; 6 mL, 52 mmol), paraformaldehyde (6.2 g, 207 mmol), concentrated hydrochloric acid (2.7 mL), and glacial acetic acid (4 mL). This mixture was stirred for 15 min under nitrogen. A solution of 1-(3-butoxypropyl)piperid-4-one (**5**; 11 g, 52 mmol) in glacial acetic acid (4 mL) and methanol (17 mL) was added dropwise to the original solution. After heating the reaction mixture for 10 h at 60–65 °C, a second equivalent of paraformaldehyde was added, and the resulting mixture was held at that temperature for another 10 h under nitrogen. The solvent was then evaporated, and the residue was dissolved in water (82 mL). Extraction of the mixture with diethyl ether gave two layers. The aqueous layer was made alkaline with NaOH. Upon cooling the mixture (at pH 12), a product was extracted with chloroform, and the extract was dried (MgSO_4_). Evaporation of the solvent gave a material which was purified by column chromatography over Al_2_O_3_ (eluent-benzene:isopropanol. = 6:1). The ketone 3-(3-butoxypropyl)-7-[3-(1*H*-imidazol-1-yl)propyl]-3,7-diazabicyclo[3.3.1]nonane-9-one (**7**) was obtained as an oil in a yield of 12.8 g (68%). The compound had an Rf 0.56 (Al_2_O_3_, eluent-benzene:isopropanol, 6:1) and an n_D_^20^ 1.5043.

Calculated for C_20_H_34_N_4_O_2_: C, 66.26; H, 9.45; N, 15.46.

Found: C, 66.28; H, 9.42; N, 15.45.

IR spectrum cm^−1^: 1734 (C=O), 1110 (C-O-C).

^1^H NMR spectrum, ppm: *bicyclononan-9-on:* 2.90 (2 H, H-1,5), 2.18–2.33 (4 H, H-2_ax_,4_ax_,6_ax_,8_ax_), 2.90–3.01 (4 H, H-2_eq_,4_eq_,6_eq_,8_eq_). *(1H-imidazol-1-yl)propyl:* 1.56 (2 H, NCH_2_CH_2_CH_2_N), 3.27, 3.89–4.04 (4 H, NCH_2_CH_2_CH_2_N); 6.80–6.84, 7.10, 7.44–7.45 (3H, CH_imidazol_),*butoxypropyl:* 0.84 (3 H, CH_3_), 1.41–1.65 (6 H, CH_2_), 2.66, 3.30–3.41 (6 H, NCH_2_CH_2_CH_2_O), CH_2_OCH_2_CH_2_CH_2_CH_3_).

^13^C NMR ppm: *bicyclononan-9-on:* 58.4 (C-1,5), 66.9–68.6 (C-6,8), 66.9–68.6 (C-2,4), 214.4 (C-9). *(1**H-imidazol-1-yl)propyl:* 28.5 (CH_2_CH_2_CH_2_),53.1 (CH_2_CH_2_CH_2_), 46.6 (CH_2_CH_2_CH_2_), 119.9, 131.3, 137.8 (CH_imidazol_), *butoxypropyl:* 14.3 (OCH_2_CH_2_CH_2_CH_3_), 19.4 (OCH_2_CH_2_CH_2_CH_3_), 27.7 (NCH_2_CH_2_CH_2_O), 31.9 (OCH_2_CH_2_CH_2_CH_3_), 53.6 (NCH_2_CH_2_CH_2_O), 63.0 (CH_2_CH_2_CH_2_O, 70.2 (OCH_2_CH_2_CH_2_CH_3_).

#### 4.1.3. Syntheses of β-CD Complexes of Bispidines (Compounds **8**–**11**)

*3-Boc-7-[3-(1H-imidazol-1-yl)propyl]-3,7-diazabicyclo[3.3.1]nonane* (**8**). To a mixture of 3-Boc-7-[3-(1*H*-imidazol-1-yl)propyl]-3,7-diazabicyclo[3.3.1]nonan-9-one (**6**; 4.0 g, 11.49 mmol) and hydrazine hydrate (1.77 mL, 57 mmol, 99% solution) in triethylene glycol (33.5 mL) at 65–70 °C, was added KOH (8 g, 143 mmol). The reaction mixture was heated to 160–170 °C and stirred at this temperature for 4 h. Water and excess hydrazine were distilled off at a temperature of 190–200 °C. After cooling the reaction mixture to room temperature, distilled water (56.6 mL) was added, and the mixture was extracted with diethyl ether. The extract was dried (MgSO_4_), and the solvent was evaporated. The resulting solid was purified by column chromatography over Al_2_O_3_ (eluent-benzene:isopropanol-7:1). The 3-Boc-7-[3-(1*H*-imidazol-1-yl)propyl]-3,7-diazabicyc-lo[3.3.1]nonane (**8**) was obtained (2 g, 52%) in the form of an oil with an Rf 0.22 (Al_2_O_3_, eluent -benzene:isopropanol = 7:1); n_D_^20^ 1.4680.

Calculated for C_18_H_30_N_4_O_2_: C, 64.64; H, 9.04; N, 16.75.

Found: C, 64.67; H, 9.01; N, 16.77.

IR spectrum cm^−1^: 1112 (C-O-C).

^1^H NMR spectrum, ppm: *bicyc**lononane:* 1.32 (2 H, H-1,5), 1.53 (2H, H-9), 2.16, 3.98–4.05 (4 H, H-2_ax_,4_ax_,6_ax_,8_ax_), 2.57–2.58, 4.56–4.67 (4 H, H-2_eq_,4_eq_,6_eq_,8_eq_*). (1H-imidazol-1-yl)propyl:* 1.55–1.60 (2 H, NCH_2_CH_2_CH_2_N), 3.05–3.17, 3.85 (4 H, NCH_2_CH_2_CH_2_N), 6.96, 7.03, 7.49 (3H, CH_imidazole_). *Boc:* 1.48 (9 H, CH_3_).

^13^C NMR spectrum, ppm: *b**icyclononane:* 31.2 (C-1,5), 35.0 (C-9), 48.8, 55.3 (C-2,4,6,8). *(1H-imidazol-1-yl)propyl:*29.4 (CH_2_CH_2_CH_2_), 50.1 (CH_2_CH_2_CH_2_), 47.7 (CH_2_CH_2_CH_2_), 120.5, 133.9, 136.3 (CH_imidazol_). *Boc*: 28.3 (CH_3_), 80.2 (O-C); 160.8 (C=O).

*Complex 3-Boc-7-[3-(1H-imidazol-1-yl)propyl]-3,7-diazabicyclo[3.3.1]nonane* (**9**). A solution was made of 3-Boc-7-[3-(1*H*-imidazol-1-yl)propyl]-3,7-diazabicyclo[3.3.1]nonane (**8**, 1.0 g, 3 mmol) in ethyl alcohol (25 mL) and β-CD (3.39 g, 3 mmol) in distilled water (40 mL). The solution was heated 8 h, and ethanol and water were evaporated at 50–55 °C. The residue 3-Boc-7-[3-(1*H*-imidazol-1-yl)-propyl]-3,7-diazabicyclo[3.3.1] nonane with β-CD (**9**) (3.89 g, 89%) was formed as a white powder.

Calculated for C_60_H_100_N_4_O_37_: C, 49.04; H, 6.81; N, 3.81.

Found: C, 49.01; H, 6.79; N, 3.84.

*3-(3-Butoxypropyl)-7-[3-(1H-imidazol-1-yl)propyl]-3,7-diazabicyclo[3.3.1]nonane* (**10**).To a mixture of 3-(3-butoxypropyl)-7-[3-(1*H*-imidazol-1-yl)propyl]-3,7-diazabicyclo[3.3.1]-nonane-9-one (**7**; 5.0 g, 13.8 mmol) and hydrazine hydrate (2.208 g, 69 mmol, 99% solution) in triethylene glycol (40 mL) at 60 °C was added KOH (9.58 g, 171 mmol). The reaction mixture was heated to 160–170 °C and stirred at this temperature for 4 h. Water and excess hydrazine were distilled off at a temperature of 190–200 °C. After cooling the reaction mixture to room temperature, 68 mL of distilled water was added. The product was extracted with diethyl ether, and the extract was dried (MgSO_4_). Evaporation of the solvent gave a product which was purified by column chromatography on Al_2_O_3_ (eluent-benzene:isopropanol = 7:1). 3-(3-Butoxypropyl)-7-[3-(1*H*-imidazol-1-yl)propyl]-3,7-diazabicyclo[3.3.1]nonane (**10**) was obtained as a yellow oil (1.93 g, 40%). The compound (Rf 0.25-eluent-benzene:isopropanol = 7:1 Al_2_O_3_, eluent-benzene: isopropanol = 7:1) was isolated and had an_D_^20^ 1.5094.

Calculated for C_20_H_36_N_4_O: C, 68.92; H, 10.41; N, 16.08.

Found: C, 68.90; H, 10.38; N, 16.06.

IR spectrum, cm^−1^: 1115 (C-O-C).

^1^H NMR spectrum, ppm: *bicyclononan-9-on*: 1.88 (2H, H-9), 1.89 (2 H, H-1,5),2.18–2.25 (4 H, H-2_ax_,4_ax_,6_ax_,8_ax_), 2.57–2.72 (4 H, H-2_eq_,4_eq_,6_eq_,8_eq_). (*1**H-imidazol-1-yl)propyl,*1.58 (2 H, NCH_2_CH_2_CH_2_N), 3.11–3.16, 3.87 (4 H, NCH_2_CH_2_CH_2_N), 6.86, 7.14, 7.61 (3H, CH_imidazol_), *butoxypropyl:*0.86 (3 H, CH_3_), 1.44–1.71 (6 H, CH_2_), 2.46, 3.36–3.38 (6 H, NCH_2_,CH_2_CH_2_O, CH_2_OCH_2_CH_2_CH_2_CH_3_).

^13^C NMR spectrum, ppm: *bicyclononane:* 27.7 (C-1,5), 35.1 (C-9), 60.7–62.7 (C-2,4,6,8). *(1H-imidazol-1-yl)propyl:*30.6. (CH_2_CH_2_CH_2_), 44.0 (CH_2_CH_2_CH_2_), 51.3 (CH_2_CH_2_CH_2_), 120.1, 129, 137.8. (CH_imidazol_),*butoxypropyl:* 14.3 (OCH_2_CH_2_CH_2_CH_3_), 19.4 (OCH_2_CH_2_CH_2_CH_3_), 27.8 (NCH_2_CH_2_CH_2_O), 31.8 (OCH_2_CH_2_CH_2_CH_3_). 52.3 (NCH_2_CH_2_CH_2_O), 66.9 (CH_2_CH_2_CH_2_O), 70.1 (OCH_2_CH_2_CH_2_CH_3_).

*Complex of 3-(3-butoxypropyl)-7-[3-(1H-imidazol-1-yl)propyl]-3,7-diazabicyclo [3.3.1]nonane* (**11**). Solutions of 3-(3-butoxypropyl)-7-[3-(1*H*-imidazol-1-yl)propyl]-3,7-diazabicyclo[3.3.1]nonane (**10**; 0.5 g, 1.4 mmol) in ethyl alcohol (25 mL) and β-CD (1.629 g, 1.4 mmol) in 40 mL of distilled water were mixed. The mixture was placed in an oven where ethanol and water were evaporated at 50–55 °C. The inclusion complex **11** of 3-(3-butoxypropyl)-7-[3-(1*H*-imidazol-1-yl)propyl]-3,7-diazabicyclo[3.3.1]nonane with β-CD was obtained in the form of a white-brown powder (1.95 g, 92%).

Calculated for C_62_H_106_N_4_O_36_: C, 50.19; H, 7.15; N, 3.78.

Found: C, 50.17; H, 7.17; N, 3.80.

#### 4.1.4. Syntheses of β-CD Complexes of O-Benzoyloxime of Bispidine (Compounds **12**–**14**)

*Oxime of 3-(3-butoxypropyl)-7-[3-(1H-imidazol-1-yl)propyl]-3,7-diazabicyclo[3.3.1]-nonan-9-one* (**12**). To a three-necked flask equipped with a mechanical stirrer, a reflux condenser with a calcium chloride tube, and a dropping funnel, was placed 3-(3-butoxypropyl)-7-[3-(1*H*-imidazol-1-yl)propyl]-3,7-diazabicyclo[3.3.1]nonan-9-one (**7**; 6 g, 16.57 mmol) in ethyl alcohol (97.5 mL) and pyridine (1.96 g 25 mmol). Stirring was initiated, and hydroxylamine hydrochloric acid (3 g, 43 mmol) was added. The reaction mixture was heated at 110–120 °C for 24 h. Evaporation of the solvent left a residue which was dissolved in water (30 mL), and the mixture was made alkaline with NaOH to pH 12. Extraction (HCCl_3_) of the aqueous solution gave a solution which was dried (MgSO4). Evaporation of the solvent gave a residue which was purified via column chromatography on Al_2_O_3_ (eluent-benzene:isopropanol = 20:1). Oxime **12** of 3-(3-butoxypropyl)-7-[3-(1*H*-imidazol-1-yl)propyl]-3,7-diazabicyclo[3.3.1]-nonane-9-one was obtained as an oil (3.09 g, 49%). Supporting date are: Rf 0.20 (Al_2_O_3_, benzene:isopropanol l = 20:1); n_D_^20^ 1.4910.

Calculated for C_20_H_35_N_5_O_2_: C, 63.63; H, 9.34; N, 18.55.

Found: C, 63.66; H, 9.31; N, 18.57.

IR spectrum, cm^−1^: 1506 (C=N), 3200 (O-H).

^1^H NMR spectrum, ppm: *bicyclononan-9-ketoxime:* 2.23–2.46 (4 H, H-2_ax_,4_ax_,6_ax_,8_ax_), 2.71–2.95 (2 H, H-1,5), 2.71–2.95 (4 H, H-2_eq_,4_eq_,6_eq_,8_eq_), 6.82 (H, N=OH). (*1H-imidazol-1-yl)propyl:* 1.53–1.60 (2 H, NCH_2_CH_2_CH_2_N), 3.23–3.30, 3.90–4.01 (4 H, NCH_2_CH_2_CH_2_N), 6.82, 7.11, 7.57 (3 H, CH_imidazol_), *butoxypropyl:* 0.78–0.83 (3H, CH_3_), 1.24–1.26, 1.38–1.40 (4 H, CH_2_), 2.23–2.46, 3.48–3.52 (6 H, NCH_2_CH_2_CH_2_OCH_2_), OCH_2_CH_2_CH_2_CH_3_).

^13^C NMR spectrum, ppm: *bicyclononan-9-ketoxime:*34.8 (C-5), 58.4 (C-6,8), 57.1 (C-2,4), 159.5 (C-9*). (1H-imidazol-1-yl)propyl:* 30.5 (CH_2_CH_2_CH_2_), 50.9 (CH_2_CH_2_CH_2_), 46.9 (CH_2_CH_2_CH_2_), 119.9, 128.6,138.0 (CH_imidazol_),*butoxypropyl:* 14.3 (OCH_2_CH_2_CH_2_CH_3_), 27.6 (NCH_2_CH_2_CH_2_O), 31.9 (OCH_2_CH_2_CH_2_CH_3_), 51.5 (NCH_2_CH_2_CH_2_O), 68.8 (NCH_2_CH_2_CH_2_O), 70.2 (OCH_2_CH_2_CH_2_CH_3_).

*O-Benzoyloxime of 3-(3-butoxypropyl)-7-[3-(1H-imidazol-1-yl)propyl]-3,7-diazabicyclo[3.3.1]nonan-9-one* (**13**). A mixture of oxime of 3-(3-butoxypropyl)-7-[3-(1*H*-imidazol-1-yl)propyl]-3,7-diazabicyclo[3.3.1]nonane-9-one (2.0 g, 5.3 mmol, **12**) in absolute benzene (42.4 mL) and benzoyl chloride (0.6 mL, 5.2 mmol) was stirred for 7 h at 80 °C. The solvent was distilled off, and the residue was treated with aqueous potash. Extraction (HCCl_3_) of the product gave a solution which was dried (MgSO_4_). Evaporation of the solvent left a residue which was purified by column chromatography on Al_2_O_3_ [eluent-benzene:isopropanol = 7:1] and gave the O-benzoyloxime of 3-(3-butoxypropyl)-7-[3-(1*H*-imidazol-1-yl)propyl]-3,7- diazabicyclo[3.3.1]nonane-9-one (**13**) (1.15 g, 45%) as an oil. Supporting data included an Rf 0.34 (Al_2_O_3_, eluent-benzene:isopropanol = 7:1) and n_D_^20^ 0.724.

Calculated for C_27_H_39_N_5_O_3_: C, 67.33; H, 8.16; N, 14.54.

Found: C, 67.34; H, 8.13; N, 14.52.

IR spectrum cm^−1^: 1723 (C=O).

^1^H NMR spectrum, ppm: *ester of bicyclononan-9-ketoxime:* 2.26–2.46 (4 H, H-2_ax_,4_ax_,6_ax_,8_ax_), 3.23–3.29 (2 H, H-1,5), 3.23–3.29 (4 H, H-2_eq_,4_eq_,6_eq_,8_eq_), 7.49–7.94 (5H, Ph). *(**1H-imidazol-1-yl)propyl:* 1.55–161 (2 H, NCH_2_CH_2_CH_2_N), 3.67–3.81, 4.25–4.38 (6 H, NCH_2_CH_2_CH_2_N), 7.35–7.36, 7.47–7.51 (3 H, CH_imidazol_),*butoxypropyl:* 0.78–0.81 (3 H, CH_3_), 1.20–1.26, 1.38 (4 H, CH_2_), 1.55–1.61, 3.67–3.81 (6 H, NCH_2_CH_2_CH_2_O, CH_2_OCH_2_CH_2_CH_2_CH_3_).

^13^C NMR spectrum, ppm, *ester of bicyclononan-9-ketoxime:* 34.5, 35.8 (C-1,5), 58.3–58.6, 61.3 (C-2,4,6,8), 169.1 (C-9). *(1H-imidazol-1-yl)propyl:* 31.8 (CH_2_CH_2_CH_2_), 46.4 (CH_2_CH_2_CH_2_), 51.3 (CH_2_CH_2_CH_2_), 119.9,134.0, 137.8 (CH_imidazol_),*butoxypropyl:* 14.3 (OCH_2_CH_2_CH_2_CH_3_), 19.4 (OCH_2_CH_2_CH_2_CH_3_), 27.4 (CH_2_CH_2_CH_2_), 31.8 (OCH_2_CH_2_CH_2_CH_3_), 54.9 (NCH_2_CH_2_CH_2_); 68.7 (CH_2_CH_2_CH_2_O), 70.1 (OCH_2_CH_2_CH_2_CH_3_), *phenyl:* 129.3–130.0 (CH), 166.1 (C=O).


*Complex of O-Benzoyloxime of 3-(3-butoxypropyl)-7-[3-(1H-imidazol-1-yl)propyl]-3,7-diazabicyclo[3.3.1]nonan-9**-**one (**14**)*


To obtain the inclusion complex, solutions of the *O*-benzoyloxime (**13**; 0.5 g, 1 mmol) in ethyl alcohol (25 mL) and β-CD (1.8 g, 1 mmol) in distilled water (30 mL) were mixed. The mixture was placed in an oven where ethanol and water were evaporated at 50–55 °C. The inclusion complex of **13** with β-CD was obtained as a white powder **14** (1.52g, 89%).

Calculated for C_69_H_109_N_5_O_38_: C, 51.27; H, 6.75; N, 4.33.

Found: C, 51.24; H, 6.72; N, 4.36.

### 4.2. Biological Experimental Part

The studies were performed on model samples of spring wheat *Triticum aestivum* with the varieties Kazakhstanskaya-10, Severyanka and Miras. Severyanka and Miras are drought tolerant varieties, and Kazakhstanskaya-10 was studied as a standard. Presowing treatment of wheat with aqueous solutions of samples at a concentration of 0.01% was carried out in Petri dishes during germination of the seeds for 2 days. The seedlings were then transferred to small floating rafts for germination in hydroponics up to 10 days. Then biometric parameters were taken in comparison with the control.

Water was used as the basis for the germination of the crops. Experimental samples were taken in 15 replicates for each culture and the tested growth regulator. As a control option, tap water was used and the seedlings were allowed to stand for three days. The experiment was conducted over 10 days. For each sample, the degree (%) of stimulation or inhibition was calculated compared to the control series in all experiments (The activity of **1**, **2**, **9**, **11** was studied in different series of experiments; therefore, Table 1 presents the control data of each series).

The best results were observed with the bispidine β-CD complexes of the ethers **2**, **9**, and **11**. With **2** (7-methoxypropyl analog) the length of the roots increased in all three wheat seedlings, but markedly so with Kazakhstanskaya-10 and Severyanka, with increases of 31–33% (see Table 1). Interestingly, stem growth in these wheat varieties was slightly inhibited, but only to the extent of 3–5%. The complex of **9** resulted in growth increase only in the Severyanaka variety with values of 0.7% and 11.5%, respectively. With the other two wheat varieties, the complex of **9** only slightly inhibited the development of stems. The inhibitory properties of sample **14** demonstrated the sensitivity of the plant growth to the nature of the substitutents in the bispidine unit. The data of the experiment were analyzed statistically using two-way analysis of variance (ANOVA), with varieties and treatments as main effects of shoots and roots length (Table 1).

## 5. Conclusions

It is clear that the protector/regulator ability of these 3,7-diazabicyclo[3.3.1]nonane systems is very sensitive to the substituents, particularly on C-9 bridge. For example, with 11 (7-butoxypropyl analog), growth was stimulated in both the stem and length of the roots in all three varieties of wheat. Surprisingly, shoot growth in the three wheat varieties was only slightly inhibited by 2 by about 3–5%. Interestingly, experiments revealed that the pre-sowing treatment of seeds with samples **1**, **2**, **9**, and **14** resulted in the plant height being reduced from 0.36% to 26.5%, and the root length decreased from 2.6% to 28.9% (**1**, **9**, **14**). Sample **14** inhibited the growth of the shoot and roots in all three wheat varieties. In Table 1, the mean and standard deviation for each sample is compared to the control. Although certain structural properties of the bicyclo[3.3.1]nonane systems are significant for useful activity, the observations demonstrated service as protectors/regulators for some varieties of wheat seeds. The results suggest the importance in aiding seed protection/growth regulation/pathogen safety, particularly in arid lands.

## Data Availability

The datasets used and/or analyzed during the present study are available from the corresponding author on reasonable request.

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
