# Peer review of "Novel Complexes of 3-[3-(1H-Imidazol-1-yl)propyl]-3,7-diaza-bispidines and β-Cyclodextrin as Coatings to Protect and Stimulate Sprouting Wheat Seeds"

_molecules, 2022, doi:10.3390/molecules27217406_

Round 1

Reviewer 1 Report

This paper describes attempts to synthesis new derivatives of bispidines and to study their interaction with cyclodextrins. The general aim behind all the studies is to develop novel seeds protection compounds. Indeed, this question is of great importance throughout the World. And one of the ways to solve it is the application of organic molecules with known crop-protective effects with solubilizers/stabilizers like cyclodextrines.

In general, the Referee has no great principal objections, but the sum of minor ones led me to the conclusion to ask the authors for the major revision of their manuscript.

The reasons for this are as follows:

1. Lines 22-24: the idea of this sentence is not clear

2. The same is true for lines 23-25

3. Line 27: why such complexes are "required"?

4. Figure between lines 58 and 59 (and also Fig. 1): I feel that "C3H6" must be written using a ChemDraw (or similar Structure Editor) to make the whole picture more "organic": I mean two lines with three corners: the same is true for "C4H9" (Fig. 1) Also - if you tell us about the inclusion complexes, please, indicate this by using the sigh "@": for example, "bispidine@CD"

5. Line 56-58: the mean of the sentence is not clear

6. Lines 64-65: the bispidine MUST contain a piperidine ring; the sentence is meaningless

7. Lines 65-67: the mention of the Institute is not necessary.

8. Lines 70-71:"A few bispidinones are known [21-24]". Actually, this is not true, since even only in my lab we have prepared dozens. Not a complete list of references could be found, for example, in these two papers published in Molecules:

Molecules – 2021 – 26(24) – 7539; DOI: http://dx.doi.org/10.3390/molecules26247539

Molecules – 2022 – 27(2) – 430; DOI: https://doi.org/10.3390/molecules27020430

9. Figure 1 must be re-written, since no any experimental data are present. tert-Butoxycarbonyl could be reduced as "BocO"

10. Line 76: why two carbonyl vibrations are detected?

11. Line 100: why a signal for H-9 is in a form of "triplet"?

12. Lines 107-109: "...oximation was initiated.... and gave,..." - must be re-written

13. Lines 134-136: how the authors made the complexes with CD?

14. Line 135: what is "nonanes"?

15. Lines 159-160: where is noun after "dramatic"?

16. Line 228, 235-236: why the inert atmosphere was needed?

17. General for the whole Experiment: I do not accept the writing of the NMR specta in part like different for "bicyclononane", for "imidazole", for "side chains"

18. Line 314: what is the time for the heating process?

19. For compounds 9, 11, 14 - there is no any proofs for the complex formation except for the data of elemental analysis. And this is the MAIN DRAWBACK of this paper!

And also some minor remarks:

1. Line 20: "new, novel" - please, leave one of this

2. Line 21: "and one containing" - what this means? I checked the number of compounds synthesized and found at least two molecules with BOC group

3. Line 36: "he" should be "the"

4. Line 119: "ppmwas"

5. Line 125: "ppmfrom"

6. Line 146, 158: "3,7-diazabicyclo[3.3.1]nonanes"

7. Line 169: "wasstimulated"

8. Line 448: "diaxas"

Author Response

Response to Reviewer 1

Point 1: Lines 22-24: the idea of this sentence is not clear

Response 1: Thank you very much for the reminder. We revised the sentence as follows:

‘We report the syntheses and characterization of novel 3,7-bicycl[3.3.1]bispidines possessing an imidazolpropyl group attached to N-3 and at N-7 a Boc group as well as a benzoylated-oximated group at C-9.  These compounds were complexed with b-cyclodextrin [b-CD] and evaluated as seed protectors of selected wheat seedlings. [P. 1, lines 19-22]

Point 2. The same is true for lines 23-25

Response 2: Thanks for your kind reminders. We revised the sentence as follows:

‘Using strong acid, condensations of N-substituted piperidones with the appropriate imidazole-propyl groups at N-3 and N-7 led to bispidinones 6 and 7. ‘ [P. 1, lines 22-24]

Point 3. Line 27: why such complexes are "required"?

Response 3: Thanks for your question. The complexes were needed to ensure aqueous solubility of the substituted bispidines. Moreover, the inner cavity of the bispidines is hydrophobic and can form inclusion compounds with other organic molecules.

Point 4. Figure between lines 58 and 59 (and also Fig. 1): I feel that "C3H6" must be written using a ChemDraw (or similar Structure Editor) to make the whole picture more "organic": I mean two lines with three corners: the same is true for "C4H9" (Fig. 1) Also - if you tell us about the inclusion complexes, please, indicate this by using the sigh "@": for example, "bispidine@CD"

Response 4: Thank you very much for the comments. We have made revisions accordingly about formulas. The authors suggest that using the word b-CD  indicate for b-cyclodextrin.

Point 5. Line 56-58: the mean of the sentence is not clear

Response 5: Thank you very much for the reminder. We have meant that to obtain complexes with β-cyclodextrin, cause of their hydrophobic ability.

Point 6. Lines 64-65: the bispidine MUST contain a piperidine ring; the sentence is meaningless

Response 6: Thanks for the reminder. There are written only3,7-diazabicyclo[3.3.1]nonanes containing a piperidine ring were initiated”.

Point 7. Lines 65-67: the mention of the Institute is not necessary.

Response 7: Thank you very much for the reminder. We have made revisions accordingly.

Point 8. Lines 70-71:"A few bispidinones are known [21-24]". Actually, this is not true, since even only in my lab we have prepared dozens. Not a complete list of references could be found, for example, in these two papers published in Molecules:

Molecules – 2021 – 26(24) – 7539; DOI: http://dx.doi.org/10.3390/molecules26247539

Molecules – 2022 – 27(2) – 430; DOI: https://doi.org/10.3390/molecules27020430

Response 8:  Thank you very much for the reminder. We have added these references too.

Point 9. Figure 1 must be re-written, since no any experimental data are present. tert-Butoxycarbonyl could be reduced as "BocO"

Response 9: Thank you very much for the reminder. We have made revisions accordingly.

Point 10. Line 76: why two carbonyl vibrations are detected?

Response 10: Thanks for your question. Because we have given for two compounds.

Point 11. Line 100: why a signal for H-9 is in a form of "triplet"?

Response 11: Thank you very much for your question. There are two hydrogen atoms on H-9 and they formed three peaks.

Point 12. Lines 107-109: "...oximation was initiated.... and gave,..." - must be re-written

Response 12: Thank you very much for the reminder. The authors feel that the meaning is quite clear.

Point 13. Lines 134-136: how the authors made the complexes with CD?

Response 13: Thanks for your question. The inner cavity of cyclodextrins is hydrophobic and is capable of forming inclusion complexes with other organic molecules in aqueous solutions. The procedures to make the complexes are in the Experimental. 

Point 14. Line 135: what is "nonanes"?

Response 14: Thanks for your question. We have meant that 3-Boc- or 3-(3-butoxypropyl)-7-[3-(1H-imidazol-1-yl)propyl]-3,7-diazabicyclo[3.3.1]nonan-9-ones.

Point 15. Lines 159-160: where is noun after "dramatic"?

Response 15: Thanks for your question. We have made revisions accordingly.

Point 16. Line 228, 235-236: why the inert atmosphere was needed?

Response 16: Thank you very much for your question. Nitrogen displaces the air and keeps the inside of the container free of oxygen.

Point 17. General for the whole Experiment: I do not accept the writing of the NMR specta in part like different for "bicyclononane", for "imidazole", for "side chains"

Response 17: Thank you very much for the reminder. They were absorbed by nearby areas. The side chains are an imidazole and propyl groups.

Point 18. Line 314: what is the time for the heating process?

Response 18: Thanks for your question. The solution was heated for 6-8 h.

Point 19. For compounds 9, 11, 14 - there is no any proofs for the complex formation except for the data of elemental analysis. And this is the MAIN DRAWBACK of this paper!

Response 19: Thank you very much for the reminder. The authors feel that the elemental analyses is perfectly good support for the structures of the complexes.  It turned out that the IR and NMR spectra of bicyclononane systems are very complex (as expected) for the complexes with β-CD.

And also some minor remarks:

Point 1. Line 20: "new, novel" - please, leave one of this

Response 1: Thanks for the reminder. We have made revisions accordingly.

Point 2. Line 21: "and one containing" - what this means? I checked the number of compounds synthesized and found at least two molecules with BOC group

Response 2: Thanks for the reminder. We have meant “the next side”.

Point 3. Line 36: "he" should be "the"

Response 3: Thanks for for the reminder. We have made revisions accordingly.

Point 4. Line 119: "ppmwas"

Response 4: Thank you very much for your comment. We have made revisions accordingly.

Point 5. Line 125: "ppmfrom"

Response 5: Thank you very much for the reminder. We have made revisions accordingly.

Point 6. Line 146, 158: "3,7-diazabicyclo[3.3.1]nonanes"

Response 6: Thanks for the comment. We have made revisions accordingly.

Point 7. Line 169: "wasstimulated"

Response 7: Thank you very much for the reminder. We have made revisions accordingly.

Point 8. Line 448: "diaxas"

Response 8: Thank you very much for your comment. We have made revisions accordingly.

Reviewer 2 Report

In this work, A. B. Kaldybayeva and co-workers report on the synthesis of new 3,7-diazabicyclo[3.3.1]nonane derivatives (9, 11, and 14) containing imidazole or Boc group. Complexes were formed by heating these compounds in ethanol with beta-CD. In the continuation of the research, the authors examined the ability of the prepared complexes as coating agents on seedlings to protect and stimulate the growth of three varieties of wheat. 

I cannot evaluate the part related to biological research because I am not an expert in that field. Therefore, my evaluation will only refer to the chemical part related to the synthesis and characterization of new compounds and their complexes.

Overall, my main points of concern center on the poor level of scientific presentation and data analysis in the manuscript. Moreover, the synthetic methods and characterization techniques are not adequately presented to ensure full reproducibility by other researchers (missing reaction conditions, analysis conditions, etc.), and the introduction does not describe the actual study conducted here. Just a few notes to the authors which should definitely be revised, - the synthetic scheme (Figure 1) is unintuitive, and the description of reactants and reaction conditions is missing. In the Materials and Methods section, the preparation of CD complexes with compounds 8 to 11 and 12 to 14 is described (according to ref. 20) however, apart from elemental analysis, no further evidence is provided that inclusion complexes with CD are indeed formed.

As such, I regret to suggest the rejection of the manuscript for publication in Molecules in its present form.

Author Response

Response to Reviewer 2

Point. ‘Overall, my main points of concern center on the poor level of scientific presentation and data analysis in the manuscript. Moreover, the synthetic methods and characterization techniques are not adequately presented to ensure full reproducibility by other researchers (missing reaction conditions, analysis conditions, etc.), and the introduction does not describe the actual study conducted here. Just a few notes to the authors which should definitely be revised, - the synthetic scheme (Figure 1) is unintuitive, and the description of reactants and reaction conditions is missing. In the Materials and Methods section, the preparation of CD complexes with compounds 8 to 11 and 12 to 14 is described (according to ref. 20) however, apart from elemental analysis, no further evidence is provided that inclusion complexes with CD are indeed formed.

 As such, I regret to suggest the rejection of the manuscript for publication in Molecules in its present form'

Response. Thank you very much for your precious time in reviewing our paper and providing valuable comments. We have tried to answer and correct the manuscript using comments Reviewer 1. The purpose of our study is to synthesize novel bicyclononane derivatives and show their potential as wheat growth and development stimulants. And we used a well-known technique - to study the biological activity of the compounds in the form of their β-cyclodextrin complexes. Previously obtained 1 and 2 were also tested as 1.β-CD and 2.β-CD. It turned out that the IR and NMR spectra of bicyclononane systems are very complex (as expected) for the complexes with β-CD.

Round 2

Reviewer 2 Report

Thank you for your response, my important remark is that the authors in this manuscript did not actually prove the formation of the CD and 3,7-diazabicyclo[3.3.1]bispidines complex.  Furthermore, they concluded that complex formation is responsible for biological activity, but this opinion they based on referring to similar experiments described in the literature. The opinion of the reviewer is that the work would have gained many times its value if the formation of the incursion complex had been experimentally proven, from a chemical point of view it is an important question.